# Towards reliable quantification of cell state velocities

**Valérie Marot-Lassauzaie**[1,2☯], **Brigitte Joanne Bouman**[1,3☯], **Fearghal Declan Donaghy**[1], **Yasmin Demerdash**[4,5,6], **Marieke Alida Gertruda Essers**[4,5,7], **Laleh Haghverdi**[1]*

**1** Berlin Institute for Medical Systems Biology, Max Delbrück Center (BIMSB-MDC) in the Helmholtz Association, Berlin, Germany, **2** Charité – Universitätsmedizin Berlin, corporate member of Freie Universität Berlin and Humboldt- Universität zu Berlin, Berlin, Germany, **3** Humboldt Universität zu Berlin, Institute for Biology, Berlin, Germany, **4** Division Inflammatory Stress in Stem Cells, German Cancer Research Center (DKFZ), Heidelberg, Germany, **5** Heidelberg Institute for Stem Cell Technology and Experimental Medicine (HI-STEM gGMBH), Heidelberg, Germany, **6** Faculty of Biosciences, University of Heidelberg, Heidelberg, Germany, **7** DKFZ-ZMBH Alliance, Heidelberg, Germany

☯ These authors contributed equally to this work.

\* laleh.haghverdi@mdc-berlin.de

## Abstract

A few years ago, it was proposed to use the simultaneous quantification of unspliced and spliced messenger RNA (mRNA) to add a temporal dimension to high-throughput snapshots of single cell RNA sequencing data. This concept can yield additional insight into the transcriptional dynamics of the biological systems under study. However, current methods for inferring cell state velocities from such data (known as RNA velocities) are afflicted by several theoretical and computational problems, hindering realistic and reliable velocity estimation. We discuss these issues and propose new solutions for addressing some of the current challenges in consistency of data processing, velocity inference and visualisation. We translate our computational conclusion in two velocity analysis tools: one detailed method $\kappa$-velo and one heuristic method eco-velo, each of which uses a different set of assumptions about the data.

**Data Availability Statement:** All analysed datasets are publicly available. The pancreatic endocrinogenesis dataset is available from the Gene Expression Omnibus (GEO) under accession

## Author summary

Single cell transcriptomics has been used to study dynamical biological processes such as cell differentiation or disease progression. An ideal study of these systems would track individual cells in time but this is not directly feasible since cells are destroyed as part of the sequencing protocol. Because of asynchronous progression of cells, single cell snapshot datasets often capture cells at different stages of progression. The challenge is to infer both the overall direction of progression (pseudotime) as well as single cell specific variations in the progression. Computational methods development for inference of the overall direction are well advanced but attempts to address the single cell level variations of the dynamics are newer. Simultaneous measurement of abundances of new (unspliced) and older (spliced) mRNA in the same single cell adds a temporal dimension to the data which can be used to infer the time derivative of single cells progression through the dynamical

GSE132188. The murine gastrulation dataset is available on the Arrayexpress database (http://www.ebi.ac.uk/arrayexpress) under accession number E-MTAB-6967. For both datasets the count matrices can be downloaded directly from the scVelo Python implementation (https://scvelo.org) v0.2.4. The raw data from the chromaffin dataset is available on GEO under accession number GSE99933. The count matrices are made available at http://velocyto.org. The count matrices of the HSPC dataset are available on our GitHub Page: https://github.com/HaghverdiLab/velocity_notebooks. This GitHub page also contains all notebooks necessary to reproduce the results reported in this paper. A python implementation of the κ-velo and eco-velo pipeline can be found at https://github.com/HaghverdiLab/velocity_package.

**Funding:** This study was supported by the Max Delbrück Center for Molecular Medicine as well as the Bundesministerium für Bildung und Forschung (BMBF) grant for 'junior consortia in systems medicine' to LH (01ZX1911B) and the Dietmar Hopp Foundation, as well as SFB873 funded by the Deutsche Forschungsgemeinschaft (DFG) to MAGE. The funders had no role in study design, data collection and analysis, decision to publish, or preparation of the manuscript.

**Competing interests:** The authors have declared that no competing interests exist.

process. State-of-the-art methods for inference of cell state velocities from RNA-seq data (also known as RNA velocity) have multiple unaddressed issues. In this manuscript, we discuss these issues and propose new solutions. In previous works, agreement of RNA velocity estimations with pseudotime has been used as validation. We show that this in itself is not proof that the method works reliably and the overall direction of progression has to be distinguished from individual cells' behaviour. We propose two new methods (one detailed and one cost efficient heuristic) for estimation and visualisation of RNA velocities and show that our methods faithfully capture the single-cell variances and overall trend on simulation. We further apply the methods to different datasets and show how the method can help us gain biological insight from real data.

This is a *PLOS Computational Biology* Methods paper.

## Introduction

Single cell transcriptomics has facilitated the study of asynchronous cellular processes such as cell differentiation in the high-dimensional gene expression space. Development of computational methods for extracting temporal information from snapshots of the system has attracted much attention in recent years. The output of these methods is typically a pseudo-temporal ordering of cells, representing their progression along the (deterministic) path of directed differentiation. However, this ordering does not reflect the intrinsic stochastic characteristics of the process and leaves several biologically interesting questions unanswered. Can cells go back along de-differentiation paths? If yes, how far and how likely is that? How strong is the stochastic component of the dynamics compared to the deterministic directed part? Answering these questions would allow quantification of cell fate plasticity in different transcriptional regions.

RNA velocity, proposed by [1] (and the corresponding package called velocyto), was a breakthrough towards obtaining a more complete description of the dynamics of cell differentiation. Simultaneous measurement of abundances of nascent unspliced and mature spliced mRNA in single cells adds a temporal dimension to the collected data which can be used to infer the temporal motion of cells in transcriptomic space. A later method, scVelo [2], further advanced the concept by solving the transcriptional dynamics of splicing kinetics and velocity inference. Other extensions included additional temporal layers of gene regulation such as protein levels [3] or chromatin accessibility [4] to the unspliced and spliced mRNA levels to extract further information on cell state dynamics. Recently, there have also been advancements in using cell state velocities to study the degree of cell plasticity [5]. For all these methods, it is important to first ensure robust and reliable estimation of single cell velocities. Ideally, the estimated velocities should capture both the overall course in the population as well as the single-cell specific (stochastic) part of the dynamics. However, reliable inference of cell state velocities is still impeded by multiple computational issues. Some weaknesses in current velocity visualisation approaches, as well as challenges in inclusion of genes with multiple dynamics, have been pointed out in [1, 2, 6, 7]. Another issue on scale invariance of gene-wise velocity components was described in more detail in [8]. Current methods either do not address this scale invariance issue or address it incompletely using unrealistic assumptions.

Moreover, there are several inconsistencies in the current methods' processing pipeline and the stochastic part of the dynamics is lost through multiple layers of data imputation and smoothing. In parallel to this study, [9] and [10] point out some of the limitations and problems of current velocity visualisation methods. [9] also suggests that, due to the highly stochastic nature of gene expression process, currently used (deterministic) approaches are insufficient and propose development of probabilistic alternatives. More recently, a variational inference method for RNA velocity estimation has also become available [11].

In this manuscript, we argue that when dealing with highly stochastic processes, deterministic approaches are only useful when talking about average velocities over specific time intervals, instead of talking about spontaneous velocities which are immeasurable in real life. We propose two different approaches for estimation and visualisation of RNA velocities. In $\kappa$-velo, we first design a processing workflow specifically adapted to downstream velocity calculations, thereby addressing problems in previously used workflows. We then solve for the gene-wise reaction rate parameters and propose an approach to relate velocity components across genes, hence resolving the scale invariance issue. We also present a new visualisation method that more faithfully represents the stochastic part of the velocities. In addition, we propose eco-velo, a heuristic method that bypasses several cumbersome, computationally costly and stochasticity killing steps used by other available methods.

A table of contents is provided in S1 Table of contents.

## Methods

### Dynamical inference

Building high-dimensional cell state velocities as vector sums of their gene-wise components (as is the current practice) requires careful handling of two major issues: ambiguity of the time scales and the relative scaling between different velocity components. In this section, we discuss current problems in state-of-the-art velocity estimation approaches and introduce our novel $\kappa$-velo and eco-velo approaches.

**The time scale over which average cell state velocities are reported.**  In the physical world, we can only measure average velocities in a given time interval $\Delta t$. As $\Delta t \rightarrow 0$ measured velocities get closer to instantaneous velocities, which are impossible to measure directly. When adding multiple velocity components one would ideally need to measure all gene-wise displacement components $\vec{\Delta x_g}$ in the same interval $\Delta t$. Mathematically:

$$\vec{V} = \frac{\sum_{g=1}^{G} \vec{\Delta x_g}}{\Delta t} \tag{1}$$

where $\vec{V}$ is the $G$-dimensional velocity vector and $\Delta t$ is the same for all genes. However, in the RNA velocity framework (even without scale invariance problem discussed in the next subsection) we use:

$$\vec{V} = \sum_{g=1}^{G} \vec{v_g}, \quad \vec{v_g} = \frac{\vec{\Delta x_g}}{\Delta t_g} \tag{2}$$

the result of which is different from Eq 1 for non-smooth expression dynamics. Using a different $\Delta t_g$ for each gene $g$, raises an immediate question: which time interval does the average cell state velocity $\vec{V}$ calculated from Eq (2) correspond to? Obscurity in the physical meaning of velocities calculated as such is more pronounced when including genes with noisy expression dynamics, e.g. bursting genes where velocities will change depending on the time scale (S1

Fig). For such genes, it would be interesting to experimentally measure velocities at multiple time scales. This could help us better understand the extent of cell fate plasticity. One would expect to see more variance in the direction of individual cells velocities reported in small time scales, whereas velocities over sufficiently large time scales would better align with the pseudo-temporal direction of differentiation.

**Scale invariance of gene-wise velocity components.** According to the RNA velocity formalism:

$$
\begin{aligned}
\frac{du_g}{dt} &= \alpha_g - \beta_g u_g \\
\frac{ds_g}{dt} &= \beta_g u_g - \gamma_g s_g = v_g
\end{aligned}
\tag{3}
$$

where $u_g$ and $s_g$ represents the number of unspliced and spliced counts for gene $g$. $\alpha_g, \beta_g, \gamma_g$ represent transcription rate, splicing rate and mature mRNA degradation rate respectively. $v_g$ represent the instantaneous velocity component of gene $g$.

Eq 3 provides a deterministic (smooth) explanation of gene transcription and splicing events, in which the kinetic rate parameters are assumed to be constant (equal to their mean value over a relatively large time interval). In absence of temporal measurements (i.e., when working with snapshot u-s counts data) the actual time scales for which the assumption of constant kinetic rates for each gene would be valid are not known. In essence, one has to work with a time-independent relation between u and s counts, which we know as the u-s phase portrait of the data. This implies that scaling $dt$ (or equivalently $t$) by $\kappa$ does not change the u-s phase portrait of a gene. This scaling factor at the left hand side denominators of Eq 3 can be absorbed to the right hand side (RHS) of the equation, suggesting that if $(\alpha_g, \beta_g, \gamma_g)$ is a solution, $(\kappa\alpha_g, \kappa\beta_g, \kappa\gamma_g)$ is also a solution for any $\kappa$. This complicates deduction of the relative scaling of different genes, as was also shown in previous studies [8]. To get a valid high-dimensional velocity vector $\vec{V}$, one needs to know the real scaling factor $\kappa_g$ for each gene:

$$
\vec{V} = \sum_{g=1}^{G} \vec{v_g} = \sum_{g=1}^{G} \kappa_g \frac{ds_g}{dt} \hat{e}_g
\tag{4}
$$

where $\hat{e}_g$ represents the unit vector for gene $g$.

To overcome the scale invariance, velocyto assumes $\kappa\beta = 1$ (i.e. same splicing rate) for all genes. scVelo assumes that the time of the observed kinetics (i.e. turning on, reaching stationary state and turning off) on the u-s phase portrait is equal for all genes (using a default constant of 20 hours). They then fit a latent time between 0 and this constant to the cells on the phase portrait of each gene, and scale the other kinetic parameters accordingly.

Having a global (i.e., gene independent for each cell) latent time would put the time scales of different genes' u-s phase portraits in perspective and resolve the scale invariance issue. However, we note that optimisation of cells' global latent time is not part of the expectation maximisation procedure in scVelo. Rather, after fitting the gene-specific parameters (including the latent times of the cells), scVelo uses a multi-step ad-hoc voting method among the fitted latent times from multiple high-likelihood genes to calculate a global latent time for the cells. This approach does not realistically address the relative scale of different genes and the full cycle time for all genes remain equal by assumption.

Instead, we suggest that a proxy for typical travel times between cell states could be used as global latent time. For example, one could use pseudotime or the cell density scaled version of it called universal time [12] as proxy for actual transition time between cell states. Here, the

accuracy of pseudotime recovery for multiple branches of typical differentiation processes would be crucial for estimation of the velocity parameters.

In $\kappa$-velo, we circumvent prior recovery of global latent times and use an equivalent per-gene approach. Here, we use the number of cells between two cell states as a proxy for the typical travel time between them on the gene specific u-s phase portraits. This approach assumes that the probability of capturing cells in a given expression state is proportional to the time cells spend in that state. In eco-velo we take a different approach, which does not decompose the gene-wise velocity components in the first place but, similarly to velocyto, relies on strongly simplifying assumptions on the kinetic rate parameters.

**First approach: $\kappa$-velo.** Our first approach recovers the full dynamics of splicing kinetics and addresses the scale invariance problem by using a proxy of travel time between cell states. In this subsection, we drop the gene-wise indices $g$ as we address the scaling factor $\kappa$ for one gene at a time.

Consider one gene with true parameters of reaction rate $\theta_{true} = (\kappa\alpha, \kappa\beta, \kappa\gamma)$. In a first step, we recover an arbitrary solution of the reaction rate parameters with $\beta = 1$, i.e $\theta = (\alpha, 1, \gamma)$, and in a second step we recover the $\kappa$ which scales this solution to its actual magnitude relative to the other genes. Below we elaborate on each of the two steps.

The analytical solutions to Eq 3 are given by: [2]

$$
\begin{aligned}
u(t) &= u_0 \exp(\beta(t - t_0)) + \alpha(1 - \exp(-\beta(t - t_0))) \\
s(t) &= s_0 \exp(-\gamma(t - t_0)) + \frac{\alpha}{\gamma}(1 - \exp(-\gamma(t - t_0))) \\
&\quad + \frac{\alpha - \beta u_0}{\gamma - \beta}(\exp(-\gamma(t - t_0)) - \exp(-\beta(t - t_0)))
\end{aligned}
\tag{5}
$$

where $t \in (t_1, \ldots, t_n)$ is the gene specific latent time assigned to each cell and $u_0 = u(t_0)$, $s_0 = s(t_0)$ are the initial conditions. Transcriptional regulation is inscribed in $\alpha$, which is set to 0 at downregulation. Cells can then either be in up- or downregulation, as encoded in the parameter $k_i$, with $k = 1$ at upregulation and $k = 0$ at downregulation. We set $(u_0, s_0) = (0, 0)$ in the upregulation phase ($k = 1$) and $(u_0, s_0) = (u(t_{switch}), s(t_{switch}))$ in the downregulation phase ($k = 0$).

We note that if $t$ was given as a global (gene independent) latent time assigned to each cell, the scale invariance problem would already be resolved. However, in practice we do not have $t$.

From the solution $u(t)$, we get $\exp(-\beta(t - t_0)) = \frac{\beta u(t) - \alpha}{\beta u_0 - \alpha}$. Therefore, $\exp(-\gamma(t - t_0)) = (\exp(-\beta(t - t_0)))^{\gamma/\beta} = \left(\frac{\beta u(t) - \alpha}{\beta u_0 - \alpha}\right)^{\gamma/\beta}$. Substituting this Eq in $s(t)$, we get the time-independent relation between unspliced and spliced counts. For $\beta = 1$ specifically, we get:

$$
s(u) = \left(s_0 - \frac{\alpha}{\gamma} + \frac{\alpha - u_0}{\gamma - 1}\right)\left(\frac{u - \alpha}{u_0 - \alpha}\right)^\gamma + \frac{u - \alpha}{\gamma - 1} + \frac{\alpha}{\gamma}
\tag{6}
$$

This is the form of a function $s(u)$ which we can directly fit to the data points in a u-s phase portrait.

In practice, there is one more amendment needed for parameter fitting. As current procedures for assignment of the sequence reads to either unspliced or spliced mRNA are biased towards spliced assignments and heavily underestimate the unspliced counts, a function of the form Eq 6 cannot approximate the data unless we upscale the measured u counts by a (gene-

specific) factor $m_g$ (see Note A in S1 Appendix). Thus instead of Eq 3 we now have:

$$m_g \frac{du_g}{dt} = \alpha_g - m_g \beta_g u_g$$

$$\frac{ds_g}{dt} = m_g \beta_g u_g - \gamma_g s_g = v_g$$

(7)

We note that, through necessity from current data qualities, scVelo also scales u, but scales u to have the same variance as s [2]. In fact scaling in this way is equivalent to setting $\kappa\gamma/\kappa\beta \approx 1$ (see Note A in S1 Appendix). We also note that upscaling u by $m_g$ is different from separate normalisation as here the counts of that gene are multiplied by the same constant for all cells, whereas a separate normalisation will affect cells differently for the same gene. With $m_g$, Eq 6 becomes:

$$s(u) = \left( s_0 - \frac{\alpha}{\gamma} + \frac{\alpha - mu_0}{\gamma - 1} \right) \left( \frac{mu - \alpha}{mu_0 - \alpha} \right)^{\gamma} + \frac{mu - \alpha}{\gamma - 1} + \frac{\alpha}{\gamma}$$

(8)

which we use for fitting to the u-s data and inference of the parameters ($\alpha$, $\gamma$, $u_{switch}$, $m$) (see Note B in S1 Appendix for the details of our expectation maximisation (EM) procedure).

Once the EM is done, we recover the time scale $\kappa$ for each gene. Let $\Delta t_{ij}$ be a measure of time that can be used to relate time between two states $i$, $j$ across genes, with $i$ before $j$ in time. Consider one gene with true parameters of reaction rate $\theta = (\kappa\alpha, \kappa\beta, \kappa\gamma)$ and recovered parameters $\theta = (\alpha, \beta, \gamma)$ and $u_i$, $u_j$ the measured unspliced counts for cells $i$, $j$. Note that for that gene, $i$, $j$ need to be in the same state of transcriptional induction or repression because the speed of genes is only measurable during transcriptional change, i.e. outside of steady-state. If the cells spend time in steady state, the change in transcriptional state will not be proportional to the distance in time, which is why we only consider cells in the same state.

Considering the time scale $\kappa$ in the solution for $u(t)$ in Eq 5, yields for two measurements from cell $i$ and $j$:

$$mu_j = mu_i \exp(-\beta\kappa\Delta t_{ij}) + \frac{\alpha}{\beta}(1 - \exp(-\beta\kappa\Delta t_{ij}))$$

(9)

Solving for $\kappa\Delta t_{ij}$ we get:

$$\kappa\Delta t_{ij} = \frac{1}{\beta} \log \frac{mu_i - \alpha/\beta}{mu_j - \alpha/\beta}$$

(10)

with $\beta = 1$, and $m$ and $\alpha$ inferred from EM. As a proxy for the true $\Delta t_{ij}$s (which we do not have) we use the number of cells that occur between the cells $i$ and $j$ calling it $d(i, j)$. The rational being that, in snapshot data, the probability of capturing cells in a specific region of the expression space is proportional to the time cells typically spend in that region. This assumption serves as a valid approximation for most single-cell datasets, but is undermined in presence of non-uniform cell proliferation and death rates as well as biased sampling of cell types (e.g. enrichment for specific cell types).

Let us call the RHS of Eq 10, $f(i, j)$. For cell pairs that are in the same transcriptional phase (i.e., the upregulation or the downregulation phase), $f(i, j)$ has a linear relation to $d(i, j)$, with the slope given by $\kappa$. However if either (or both) cells are in steady-state, $f$ will be smaller than expected from Eq 10. Thus, plotting $f(i, j)$ versus $d(i, j)$ for random pairs of $i$, $j$, produces a parallelogram of which the left slope equals $\kappa$. To recover $\kappa$, we fit a parallelogram to the data points with minimum area, while maximising the number of points in the parallelogram (Note C in S1 Appendix and S2 Fig).

Here, we inferred $\kappa$ from the unspliced counts data. One could similarly use the spliced counts data and infer $\kappa$ from the $s(t)$ solution in Eq 5, which yields $\kappa$ estimations congruent with those inferred from the unspliced data (see Note D in S1 Appendix and S3 Fig). However, as $u(t)$ depends only on $\alpha$, $\beta$ while $s(t)$ depends on $\alpha$, $\beta$, $\gamma$, i.e. on one more imputed parameter, we consider recovery of $\kappa$ values from $u(t)$ as more straightforward and less error-prone.

After determination of the gene-wise $\kappa$, we are ready to call the high-dimensional, correctly scaled parameters $\Theta = (A, B, \Gamma)$, with $A_g = \kappa_g \alpha_g$, $B_g = \kappa_g \beta_g$ and $\Gamma_g = \kappa_g \gamma_g$. We call the high-dimensional unspliced counts scaling parameters $m_g$, $M$. For calculating the high-dimensional velocity for cell $i$, we thus use $\vec{V}_i = BMU_i - \Gamma S_i$, where $U_i$ and $S_i$ respectively represent the G-dimensional u-s counts in cell $i$ ($G$ being the number of genes).

**Second approach: Eco-velo.** Our second approach eco-velo estimates cell state velocities directly in the high-dimensional gene space by calculating the displacement for each cell in a fixed time interval. This approach eliminates the need for cumbersome and error-prone gene-wise parameter estimations. It also specifies the time interval over which high-dimensional velocities are reported, a feature that the gene-wise parameter estimation approaches (including $\kappa$-velo) are missing. Specification of the velocity estimation time interval can be important for data sets that include multiple non-smooth-dynamics genes where short-term cell velocities can deviate significantly from their long-term velocity directions.

Starting from Eq 3, for the change of the spliced counts of gene g over $\Delta t$ we can write:

$$
\begin{aligned}
v_g &= \beta_g u_g(t) - \gamma_g s_g(t) \Rightarrow \\
s_g(t + \Delta t) &= s_g(t) + v_g \Delta t \\
&= s_g(t)(1 - \gamma_g \Delta t) + \beta_g u_g(t) \Delta t
\end{aligned}
\tag{11}
$$

By fixing $\Delta t_g = 1/\gamma_g$ (this is the time in which existing spliced reads for gene $g$ will be degraded) we get:

$$
s_g(t + \Delta t_g) = \beta_g u_g(t) \Delta t_g = \frac{\beta_g}{\gamma_g} u_g(t)
\tag{12}
$$

This means that knowing $\beta_g/\gamma_g$ is sufficient to estimate the cell state displacements over $\Delta t_g$. If we further assume all genes have a similar $\beta$ and $\gamma$, we can conclude that the unspliced counts $u(t)$ in a cell are proportional (with a constant factor $\beta/\gamma$) to its spliced counts at the later time point $(t + 1/\gamma)$.

The assumption of similar $\gamma$ as well as $\beta$ across genes, allows us to avoid decomposition of high-dimensional velocities into gene-wise components for velocity estimation and recombining the estimated components again. Thus, leading to another level of simplification that turns out very handy as a heuristic velocity estimation from u and s counts, where we can find cell state displacements by mapping U to S. We do so by searching for the nearest neighbors (NNs) of U in S that are also within the first $k$ nearest neighbors of S in U. We call these pairs mutual nearest neighbors (MNNs). Note that not every point needs to have MNNs. The velocity arrow then goes from a cell's position in S space to the the mean of the first k MNNs of that same cell's U space in S. Here, u and s counts can be used directly for estimating cell state velocity directions without any need for smoothing and parameter fitting.

The strong assumptions of eco-velo (similar $\gamma$ and $\beta$ across genes) may not hold for every biological processes and every subset of genes. Thus here, one would ideally select a set of genes that are only transcriptionally regulated (via $\alpha$), but not post-transcriptionally regulated (involving gene-specific $\beta$ and $\gamma$ rates). An example of such a cases seems to occur in Fig 1E of the original RNA velocity paper [1], where the authors observed for bulk RNA-seq

measurements of cell cycle genes in the mouse liver over a time course of the circadian cycle, that unspliced mRNAs appear predictive of spliced mRNA at the next time point with a similar signal intensity coefficient. Conditioned on its assumptions, eco-velo (in contrast to the methods based on gene-wise parameter estimation) specifies the time interval of the reported velocities and also skips several error-prone parameter estimation and data smoothing steps. How much the different assumptions of each method are satisfied for different experimental settings, data qualities, as well as the purposes of velocity analysis (e.g. estimating the overall velocity directions or obtaining the average cell state velocities over a specific time scale) would determine which method is more appropriate to use.

## Visualisation

La Manno et al. [1] suggested using projection of the end of the velocity vectors ($\vec{s} + \vec{v}\Delta t$) with $\Delta t = 1$ on an embedding of the spliced counts. While projection using principal component analysis (PCA) (Note E in S1 Appendix) is the most accurate low-dimensional representation of cell state velocities, it usually does not capture the full complexity of differentiation manifolds with several subpopulations in high-dimensional gene space. Projection of the velocities onto non-parametric nonlinear embeddings (which do not have gene-defined axes) is more challenging. To work around this difficulty, velocyto projects the velocities in a direction relative to the neighbouring cells. This is done by computing a transition probability matrix $P$ containing probabilities of cell-to-cell transitions in accordance with the velocity vector:

$P_{ij} = \exp\left(\frac{\mathrm{corr}(\rho(\vec{s_j - s_i}), \rho(\vec{v_i}))}{\sigma^2}\right)$ with $\sigma$ the kernel width parameter, $\rho(x) = \mathrm{sgn}(x)\sqrt{|x|}$ a variance-stabilising transformation and corr() the Pearson correlation coefficient. The matrix is row-normalised so that $\sum_j P_{ij} = 1$. Given $n$ observations and $Y_i$ the positions of cell $i$ on a $K$-dimensional embedding, the projected end of velocity vector for cell $i$ is calculated as $\vec{Y_i} + \Delta\vec{Y_i}$, where:

$$\Delta\vec{Y_i} = \sum_j \left(P_{ij} - \frac{1}{n}\right)\frac{\overrightarrow{Y_j - Y_i}}{\|Y_j - Y_i\|} \tag{13}$$

To project the velocities, scVelo uses a similar approach to velocyto but with a slightly different $P$ matrix that calculates Pearson correlations (also called cosine similarity) directly on the $\vec{\Delta s}_{ij}$ and $\vec{v_i}$ vectors without using the $\rho(x)$ transformation, via $P_{ij} = \exp\left(\frac{\cos\angle(\vec{s_j - s_i}, \vec{v_i})}{\sigma^2}\right)$. A vector summation as proposed in Eq 13 used in velocyto and scVelo is questionable for three reasons. First, this approach is not faithful to the velocity vectors length, e.g., two velocity vectors with the same direction, but different length (in the same neighbourhood) in the high-dimensional space will be visualised with similar lengths. That is because they will be assigned the same $P_{ij}$ as Pearson correlation does not respect the length of the vectors. Second, $\frac{\overrightarrow{Y_j - Y_i}}{\|Y_j - Y_i\|}$ does not in general provide an orthonormal basis as the direction of several neighbouring cells to cell $i$ can be correlated on the low dimensional embedding. As a result, this approach may change the direction of the velocity vectors depending on how much the orthonormality principle is disturbed for a given neighbourhood. For example, if the chosen neighbourhood extends longer along the differentiation path than its width, velocities will be visualised as more smooth vectors along the path. Third, $\left(P_{ij} - \frac{1}{n}\right)$ can be negative even if the velocity direction $\vec{v_i}$ is correlated with the direction of a neighbouring cell $j$, which is not correct.

**Nyström projection (velocity visualisation for $\kappa$-velo).** To deal with visualisation of complex data manifolds which require nonlinear embeddings, in $\kappa$-velo we propose using the Nyström projection which is more faithful to the actual high-dimensional estimated cell state velocities than the current practices. We use a nonlinear visualisation of the (normalised) spliced counts of the single cells as the start of the velocity vectors and project the end points of the velocity vectors onto this existing embedding using the Nyström method. Nyström projection has also been used for other single cell data integration applications e.g. in [13, 14]. The nonlinear embedding choice is arbitrary and can be diffusion maps [15], t-distributed stochastic neighbor embedding (t-SNE) [16] and uniform manifold approximation and projection (UMAP) [17].

If a $K$ dimensional embedding $Y_{train}$ has been created for $n_{train}$ data points $X_{train}$ and we want to project a set of $n_{test}$ points $X_{test}$ on the existing map, we first compute a transition probability matrix between the new and old data points, $P'$ of size $[n_{test}, n_{train}]$ calculated as:

$$
\begin{aligned}
Z(i) &= \sum_{j=1}^{n_{train}} \exp\left(-\frac{\|x_i - x_j\|^2}{2\sigma_i^2}\right), \quad x_i, x_j \in X_{train} \\
P'(i,j) &= \frac{1}{Z(i)} \exp\left(-\frac{\|x_i - x_j\|^2}{2\sigma_i^2}\right), \quad x_i \in X_{train}, \ x_j \in X_{test}
\end{aligned}
\tag{14}
$$

Note that when the test data is exactly the same as the training set $X_{train} = X_{test}$, $P'$ would (ideally) be the same transition matrix as the one used for generation of the train set embedding (ideally one would use the same parameters $\sigma_i$ as used in construction of the transition matrix for generating the train set embedding. See Note F in S1 Appendix for the spacial case of projection on Diffusion maps). The projection of new points $Y_{test}$ is then given by:

$$
Y_{test} = P'_{[n_{test} \times n_{train}]} * Y_{train}
\tag{15}
$$

In our application, $n_{train}$ equals $n_{test}$ as each velocity vector has a start as well as an end point.

For cell $i$ Eq 15 implies:

$$
Y_{test}(i,k) = \sum_j P'(i,j) * Y_{train}(j,k)
$$
$$
i \in \{1, .., n_{test}\}, \ j \in \{1, .., n_{train}\}, \ k \in \{1, .., K\}
\tag{16}
$$

This looks to some extent similar in form to the previously used velocity projection methods in Eq 13. However, one major difference being that we are calculating the end of the velocity arrow on the embedding space rather than the displacements, hence avoiding the collapse of velocity vectors with different lengths onto the same visualised length. Another advantage is that here $Y_{train}(j, k)$ more likely presents an orthonormal basis considering all data points, hence less affected by the shape of neighbourhoods arbitrarily chosen independent from the generation of the reference embedding. For some embedding methods (generally those which perform an analytical embedding optimisation, in contrast to the methods using iterative optimisation techniques such as gradient descent) like diffusion maps, the embedding $Y_{train}(j, k)$ is indeed guaranteed to be orthonormal (i.e., $\sum_j Y_{train}^2(j, k) = 1$ and $\Sigma_j Y_{train}(j, k_l) * Y_{train}(j, k_m) = 0$ for $k_l \neq k_m$). Lastly, all terms in $P'$ are positive, making the projected point a weighted average of the data points in the train set.

Note that the Nyström theorem is only valid for projection of test data points which are close enough to the data points existing in the training set. That is, extrapolation for test data to expression regions which have not been sampled in the training set is not possible. In $\kappa$-

velo, we ensure closeness of the end point of the velocity vector to the existing data manifold of spliced counts by adequately down scaling all inferred high-dimensional velocities by the same factor.

In light of the above, linear projection, e.g. by PCA (Note E in S1 Appendix) although not capable to capture the complexity of several datasets which consist of multiple branching events and subpopulations, remains the only approach in which the visualised arrows are a true representation of the high-dimensional velocity vectors. None of the non-parametric non-linear projection approaches can deal with projection of out of distribution data points, implying that near the boundaries of the differentiation paths, where actual velocities may point to directions going out of the existing manifold of the start point of velocity vectors, velocity visualisations will be less reliable. Moreover, embedding methods which may not keep the continuity of the data manifold (e.g. t-SNE and UMAP) are more prone to the artefacts of out of distribution data points projection.

Even though our non-linear projection method does not explicitly depend on the dimension of the train and test data sets, we recommend to use the same gene space for projecting the velocities (i.e., for computing of $P'_{[n_{test} \times n_{train}]}$) as the gene space that was used for generating the trained embedding, i.e., we use the spliced counts matrix of the filtered gene set $S$ as the training data in $\kappa$-velo. This ensures that the embedding only represents a space that can be spanned by following the velocity directions, thus making a closed set of the embedding under addition by velocity vectors. Therefore, we calculate the embedding on the same space used for parameter recovery and velocities' estimation. This also means that if we use imputed counts for parameter recovery, we calculate the low-dimensional embedding on those imputed counts.

**Visualisation for eco-velo.** For eco-velo, visualisation of velocities is integrated within the inference of the velocities and hence does not require visualisation by projection. We identify the first $k$ mutual nearest neighbours (MNNs) [18] of U and S for every cell, which we use to visualise the velocities on a low-dimensional embedding of the spliced counts. We simply draw an arrow starting from the position of a cell on the embedding to the mean of the coordinates of its first $k$ MNNs on the same embedding. That means that our velocity arrows point from $s_i$ to $\sum_j^k s_j / k$, where these $s_j$ are the first k MNNs of $u_i$ for cell $i$. These arrows corresponding to a relatively large $\Delta t$ in which all current spliced counts in the cell would be degraded. For ease of visualisation and obtaining an un-cramped map without intersecting cell velocities, we then scale all velocities by the same factor so that the arrows only point in the direction of the point and not all the way to the future state.

## Processing

Before calculating the velocities, single-cell RNAseq datasets are preprocessed (aligning the reads and counting numbers of unspliced and spliced reads) and subsequently processed (filtering, normalisation, etc.). Both the $\kappa$-velo and the eco-velo workflows start with processing raw U and S count matrices. Since the methods are based on different assumptions, the processing steps differ per method. Below, we will describe the processing protocol for both approaches.

**Processing pipeline of $\kappa$-velo.** To reduce the number of dimensions of the dataset, we select only genes with high variability. Variability is calculated on the spliced counts using analytic Pearson residuals [19]. We then filter genes with extremely low u or s counts because we want to focus only on genes with significant velocity signal. After gene filtering, the counts in each cell are size-normalised. Since the size of a cell is represented by its u and s counts together, the counts in each cell are normalised using the sum of the counts for u and s. To

recover the dynamics, the noise in the u and s counts has to be reduced. As such, all counts are imputed by averaging the counts of each cell's nearest neighbours. The nearest neighbours for each cell are found in PCA space calculated on scaled s counts. For a more detailed description of each step see Note G in S1 Appendix and S4 Fig.

**Processing pipeline of eco-velo.** Similar to $\kappa$-velo processing, the eco-velo workflow starts by filtering the dataset for genes with high variability and sufficient u and s counts. After this, all non-zero counts are log-transformed and both count matrices are normalised separately. Here, we deviate from the $\kappa$-velo protocol, because u and s counts are treated as separate modalities. Following standard MNN protocols [18], the counts are L2 normalised.

### Overview of the workflow for $\kappa$-velo and eco-velo

Both the $\kappa$-velo and the eco-velo workflows consist of three main steps: processing, velocity calculation and visualisation (Fig 1). First the data is processed as described in Section

**Fig 1. An overview of RNA velocity analysis steps in the $\kappa$-velo and eco-velo workflow.**

"Processing". In $\kappa$-velo, after processing, we recover the scaled parameters $\alpha\kappa$, $\beta\kappa$ and $\gamma\kappa$ for all genes in the dataset. For downstream velocity analysis, only genes with a likelihood above a certain threshold are used. All other genes are filtered out to reduce the technical noise caused by poorly recovered or noisy genes. Additionally, the user is provided with an option to remove genes where the order of clusters in the recovered dynamics do not match the known hierarchy of the cell types (e.g. when an assigned upregulation starts at the the most differentiated cells and ends in the progenitor population). Using the scaled parameters, a high-dimensional velocity vector is calculated for each cell. To visualise the cells and velocities, we compute an embedding (e.g. PCA, UMAP) using the processed (i.e. filtered, normalised and imputed) and scaled s counts. Lastly, the velocities are projected onto the embedding.

The eco-velo workflow includes fewer steps. After processing, the u counts are used to find the first five mutual nearest neighbours of each cell in S space. The embedding is calculated using processed (i.e. filtered and normalised) s counts and velocities are projected onto the embedding by averaging the position of the cell's first five mutual nearest neighbours.

### Simulation data

For the simulation, we randomly sampled $g$ log-normally distributed parameters of reaction rates, scaled by a scaling factor $\kappa$: $\theta = (\kappa\alpha, \kappa\beta, \kappa\gamma)$. The true time of the $n$ observations is sampled from a uniform distribution. The time points are such that the final mature steady cell state, for which all genes would reach steady-state, has not been sampled. The u and s counts are simulated following $u(t)$, $s(t)$ with added random normal noise (Note H in S1 Appendix). We simulate the data such that the time of activation of each gene's transcription is inversely proportional to the gene's speed. This means that the fastest genes are only active towards the end of the differentiation trajectory. The resulting differentiation trajectory has high velocity variation at the beginning when most genes are not yet committed to change and more deterministic dynamics with higher speed at the end of the trajectory. The motivation for this simulation scenario is to include regions with both high- and low variance velocities and to have velocities for some cells pointing to future states outside of the space observed in the original set. See Note H in S1 Appendix for a more detailed description.

### Real data

We demonstrate the performance of $\kappa$-velo and eco-velo on four different datasets and compare them with the state-of-the-art scVelo. The first dataset is a subset of the pancreatic endocrinogenesis dataset [20]. The second is a subset of the murine gastrulation dataset [21]. Both datasets were obtained using the 10x genomics platform. The third dataset consists of mouse Schwann cell precursors (SCPs) differentiating into chromaffin cells, obtained using SMART-seq2 [22]. For these three datasets, our RNA velocity analysis starts from the U and S count matrices, which were originally analysed in [2] (pancreatic endocrinogenesis), [6] (murine gastrulation) and [1] (chromaffin cells). Lastly, we also analyse a dataset of murine hematopoiesis [23], obtained using the 10x genomics platform. We used velocyto's sequence alignment and u-s counting pipeline to get the U and S count matrices, as this dataset has not been analysed for RNA velocity before. For all four datasets, we ran the complete $\kappa$-velo and eco-velo workflow as described in Section "Overview of the workflow for $\kappa$-velo and eco-velo". See Note I in S1 Appendix for further details of parameter and threshold settings for each dataset.

### Results

In this section, we first demonstrate the artefacts of scVelo's velocity projection on simulation data with known cell state velocities (i.e. no velocity inference step involved) and compare

scVelo to visualisation with linear and nonlinear projection methods. We then compare our velocities with the velocities returned by scVelo on simulation. Afterwards, we show computational experiments on real data which support the design of the processing steps we propose and use in this manuscript. In the last section we apply $\kappa$-velo and eco-velo on real datasets: first a pancreas endocrinogenesis dataset and then a hematopoiesis dataset. To validate the method on different sequencing technologies, we also applied it to a dataset of Schwann cell precursors (SCPs) differentiating into chromaffin cells ($\kappa$-velo: S5A and S5B Fig and eco-velo: S5C Fig).

## PCA and Nyström projection faithfully represent the high-dimensional velocity vectors

Ideally, a visualisation of cell state velocities should faithfully represent all aspects of the high-dimensional vector. The visualisation should respect the direction of velocity vectors as well as their magnitude (speed of change). This can be particularly difficult if the new states are in gene space not yet observed in the original set, e.g. the velocities point further than existing points. The embedding should also preserve local variations, representing fluctuations of the dynamics and cell plasticities. To assess these points, we compare existing RNA velocity visualisation methods with ours, on simulated data where the true high-dimensional velocities are known and do not need to be inferred. We design a simulation to assess all these aspects of the projection. In that simulation the cells follow a hidden true time with a high variance at the beginning and faster transitions towards the end of the trajectory (Section "Simulation data"). The final stable cell state is not yet reached in our simulation, and the velocities of the latest cells point towards not yet observed future states. Projection of the velocities on a PCA embedding (Fig 2A) reliably represents all these aspects. scVelo's velocity projection on the same PCA embedding (Fig 2B) smooths over the biologically interesting variation and removes the information on speed of change (i.e, disproportionately changes the length of the velocity vectors). The Nyström projection method (Fig 2C) captures the expected cell to cell variation, as well as the direction and length of the simulated velocities on PCA (Fig 2D). The velocities are also well represented when projected on a non-linear embedding such as t-SNE (Fig 2E and S6A and S6B Fig, UMAP shown in S6C and S6D Fig). t-SNE tends to map regions of higher density, e.g. of slower velocities, in gene space to a larger space in the embedding as highlighted by the cells outlined in blue and red. Consequently, the velocity arrows are also visualised in a scale proportional to the distance of cells in a given region on the embedding, hence looking longer than their true length in gene space. On embeddings that do not distort cell to cell distances in the gene space such as PCA or diffusion maps with a constant kernel width, the length of the velocity arrows are well represented by Nyström projection (S7 Fig diffusion map and Fig 2D PCA). We note that, unlike PCA projection, neither scVelo's nor Nyström projection are able to project end of velocity arrows that are out of distribution of existing data points.

## $\kappa$-velo recovers simulated velocities

To ensure that the high-dimensional velocity vector points in the right direction we need to address the scale invariance of gene-wise velocity components (as discussed in Section "Dynamical inference", Fig 3A). We introduce $\kappa$-velo, a method that recovers the full transcriptional dynamics from s as a function of u and thus does not need to fit a hidden latent time to the cells. The method then uses the cell densities as a proxy of time spent in a specific region of the expression space (Fig 3B) to relate velocities across genes and solve the scale invariance issue. To validate our method we simulate reaction kinetics following randomly sampled parameters scaled by a factor $\kappa$ varied between 1 and 15. The method recovers the

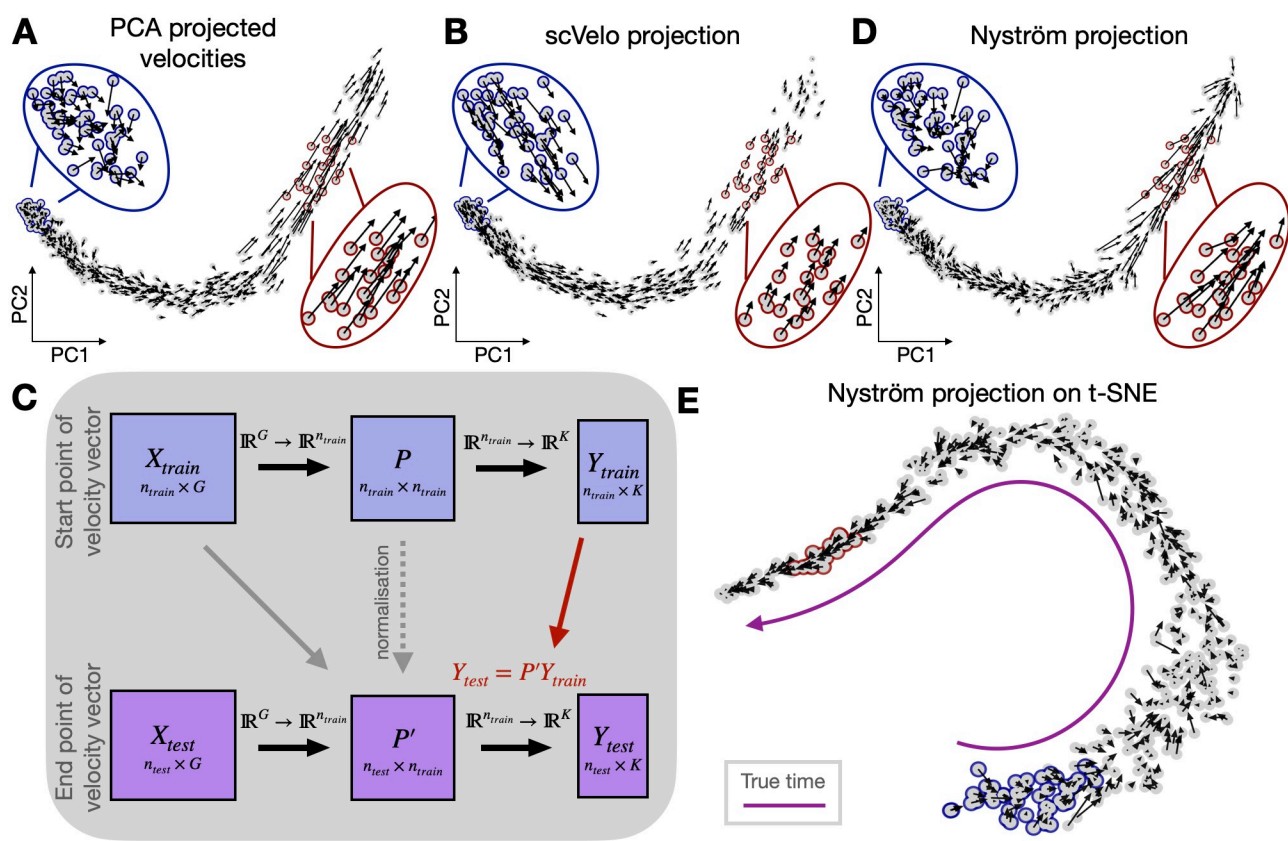

**Fig 2. Visualisation of simulated velocities with linear and nonlinear projection methods. A**. Velocities projected on PCA embedding. The blue outline highlights a region of high velocity variation and the red outline shows a low-variance, high-velocity region. The arrows in the PCA linear projection capture both the plasticity in direction and magnitude of the velocities. **B**. Velocities projected on PCA embedding by scVelo. scVelo smoothes the velocities as artefact of the projection method, thereby losing the information on cell state velocities variation as illustrated in the cells outlined in blue. scVelo also loses the information of vector length as shown in the cells outlined in red. **C**. Summary of velocity projection using the Nyström method **D-E**. Velocities projected by Nyström-projection method shown on PCA in (D) and t-SNE in (E).

scaling factors (Fig 3C). In fact, using cell densities to infer the scaling factors is equivalent to using true time for a given differentiation branch (S8 Fig). Note that the recovery becomes more difficult for higher $\kappa$. Very fast genes have few or no cells in transient state so in those cases we would need to sample more cells to reliably recover $\kappa$. We note that the scale of recovered $\kappa$ and true $\kappa$ is still off by a constant factor related to the chosen $\Delta t$, but if all components are scaled by the same factor, the direction of the high dimensional vector is still correct. After scaling, the high dimensional $\kappa$-velo velocity vector is much closer to truth (Fig 3D and S9 Fig), than scVelo's velocity vector. In fact, the errors in the scVelo vectors are proportional to the relative scale of the genes (Fig 3D). Because the high-dimensional vector is not directly conceivable to the human mind, low-dimensional representations of the velocities are usually used for interpretation of the result. We also compare the vectors after projection on a PCA embedding (S10 Fig) and find that they are also much closer to truth, both for direction and length (Fig 3E and S11 Fig). Here, for both $\kappa$-velo and scVelo, we find the biggest errors in regions of lowest and highest velocities, but scVelo's errors are much higher than $\kappa$-velo's.

## Careful processing prevents introduction of artefacts

To illustrate the importance of processing, we apply our processing pipeline to a dataset of erythroid development during murine gastrulation. Previously, it has been shown that scVelo

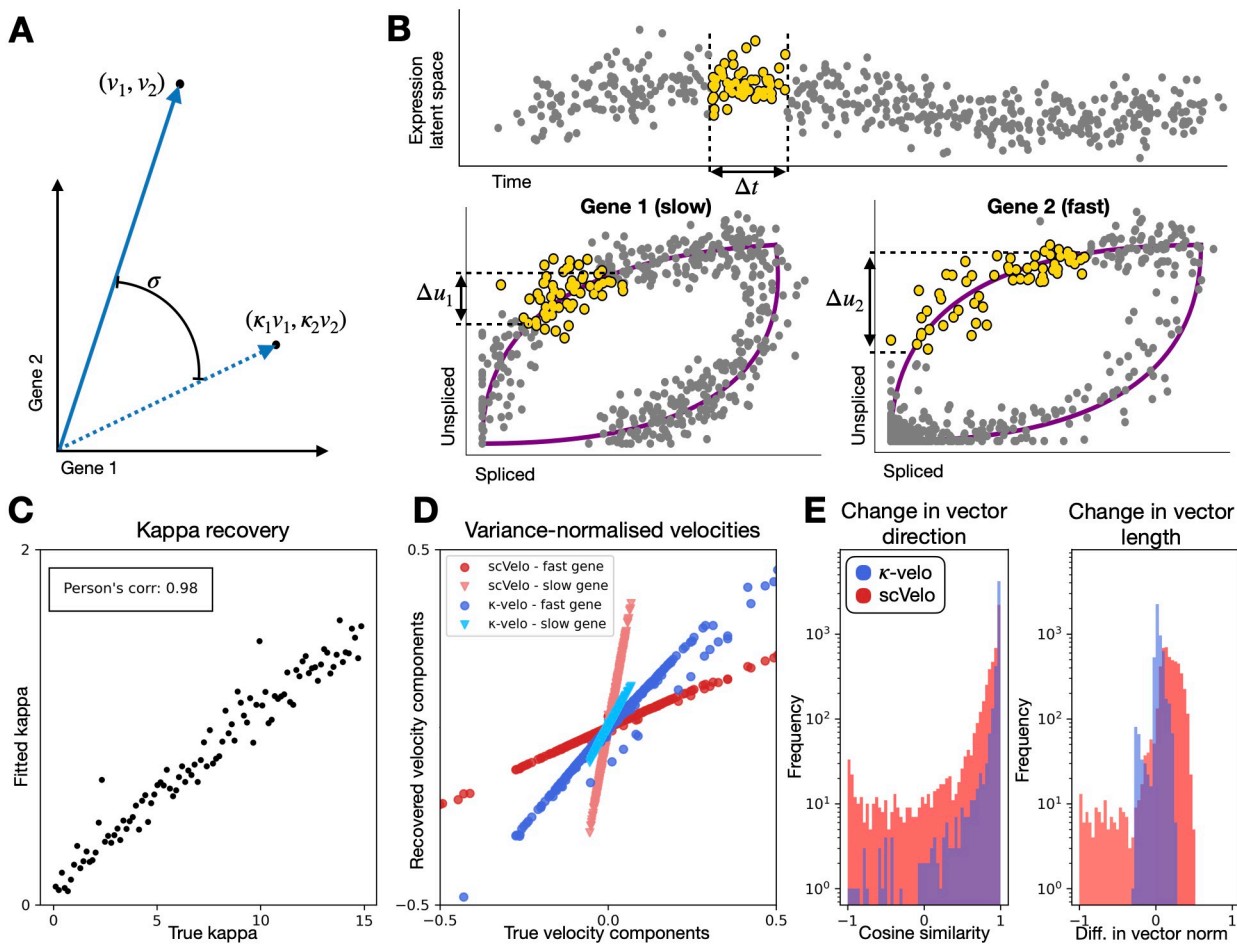

**Fig 3. Scaling of gene-wise velocity components. A**. If the gene-wise velocities are incorrectly scaled the high-dimensional velocity vector will change direction (displacement angle $\theta$). **B**. We propose to use cell densities as a proxy of time. For a same time interval, the displacement in u will be proportional to a gene's speed. This allows us to relate velocities across genes and solve the scale invariance problem. **C**. To validate $\kappa$-velo, we simulate splicing kinetics scaled by a scaling factor $\kappa$ and evaluate how well the factors are recovered. **D**. We compare the $\kappa$-velo and scVelo velocities to the true velocities for two genes with different speeds. The high-dimensional velocity vectors are normalised to have equal variance for ease of comparison. **E**. The high-dimensional vector is projected on the first two principal components to evaluate differences between true velocities and recovered velocities. We return the change in direction (cosine similarity) and length (difference in vector norm) (Note J in S1 Appendix) for $\kappa$-velo and scVelo. To make the length comparable, the vectors are variance-normalised. Note the log-scale for frequency.

falsely predicts de-differentiation at the end of erythroid development. This has been attributed to the contribution of genes with multiple rate kinetics (MURK genes) to the velocity calculation [6]. In our processing pipeline, we not only filter for low variability genes, but also remove genes with insufficient u and s counts. After normalisation, the counts are imputed by averaging spliced or unspliced counts across neighbouring cells, thereby smoothing the data. This usually produces unreliable results for genes with only few u or s counts (S9 Fig). After filtering, in scVelo's processing pipeline, the count matrices are normalised separately. This separate normalisation introduces artefacts in the u-s phase portrait (S13A and S13B Fig), which can be traced back to variation in the ratio between total unspliced and total spliced counts between cell types. We found that some of the patterns identifying MURK genes were artefacts of this normalisation (S13A and S13B Fig). Furthermore, many MURK genes in the original publication were imputed from very low counts and are filtered out in our pipeline.

Comparing the original processing pipeline to our processing steps, we reduce the number of MURK genes from 98 to 18 (S13C Fig), correcting most of the false de-differentiation.

After recovery of the parameters, we remove low-likelihood genes where the learned parameters do not fit the u-s phase portrait well. This prevents us from including the (usually noisy) genes for which the recovered parameters could be incorrect (S4 Fig: step 5). The calculated velocities for those genes would therefore not accurately reflect true dynamics. Even after filtering of low-likelihood genes, we still find genes where the recovered dynamics do not match the known order of cell types. For example, early upregulation or late downregulation can often not be easily differentiated based on the u-s phase portrait alone (S4 Fig: step 6). This could ultimately lead to incorrect velocity assignments. To avoid this issue, we can use prior information about the temporal order of cell types to perform one more round of filtering if that information is given (see Note G in S1 Appendix: step 6). We use this information to exclude genes where the fitted state assignments of up- or downregulation do not fit the expected state assignments. After both filtering steps, we calculate the low-dimensional embedding on the reduced gene set, so that the embedding only represents space that can be reached by velocities.

### $\kappa$-velo explains cell state plasticities and speed of transcriptional change in pancreas endocrinogenesis

To test whether $\kappa$-velo's velocity estimations better capture the different time scales of genes, we apply our method to a dataset of developing mouse pancreas cells sampled at embryonic day 15.5 [20]. The endocrine progenitor cells differentiate into four main fates: alpha, beta, delta and epsilon cells. In previous work, scVelo delineated cycling progenitors and the endocrine cell differentiation.

After processing, we recover the reaction rate parameters fitted by scVelo and $\kappa$-velo. True splicing rates are difficult to determine and different ranges have been reported [24] but none come close to the more than 10000-fold range reported by scVelo (Fig 4A and S14 Fig). We report a range of splicing rates close to 30-fold (Fig 4A), which is more in line with the reported ranges. After scaling, we can distinguish fast and slow genes based on their $\kappa\beta$. Among the fast genes, we find genes associated with the cell cycle such as *Adk*, while slow genes are constantly up- or downregulated during the whole differentiation trajectory (Fig 4B). This is consistent with prior expectation as the cell cycle in developing mouse pancreas takes less than a day [25], while pancreatic endocrine cell differentiation starts at embryonic day 9 and goes until day 15.5 in the analysed sample. We also find fast genes that are upregulated during commitment to a cell fate at the end of the differentiation trajectory, such as *Gcg* and *Nnat*. We note that when filtering genes based on prior knowledge of the expected order of cell types, we also filter many cycling genes that tend to have high variance, and thus partially incorrect state assignments.

We display the high dimensional vector field in a UMAP embedding of the data and compare the $\kappa$-velo velocities (Fig 4C) to scVelo velocities (Fig 4D), both projected with Nyström projection to compare only the velocity vectors (S15 Fig show projections of the velocities on a PCA embedding, S16 Fig shows smoothed velocities on the UMAP embedding). The $\kappa$-velo velocities better capture the differences in speed along the trajectory, as well as the progression within the four terminal states. scVelo's embedding (Fig 4E) smooths over the velocities, returning a view that partially appears more consistent with the expected direction of differentiation but not with the actual noisy velocity vectors. Comparing the projected velocities of the full scVelo pipeline (Fig 4E) to $\kappa$-velo pipeline (Fig 4C), we see that the methods most strongly disagree in the high-plasticity ductal population (Fig 4F and 4G and S17 Fig). There is also a

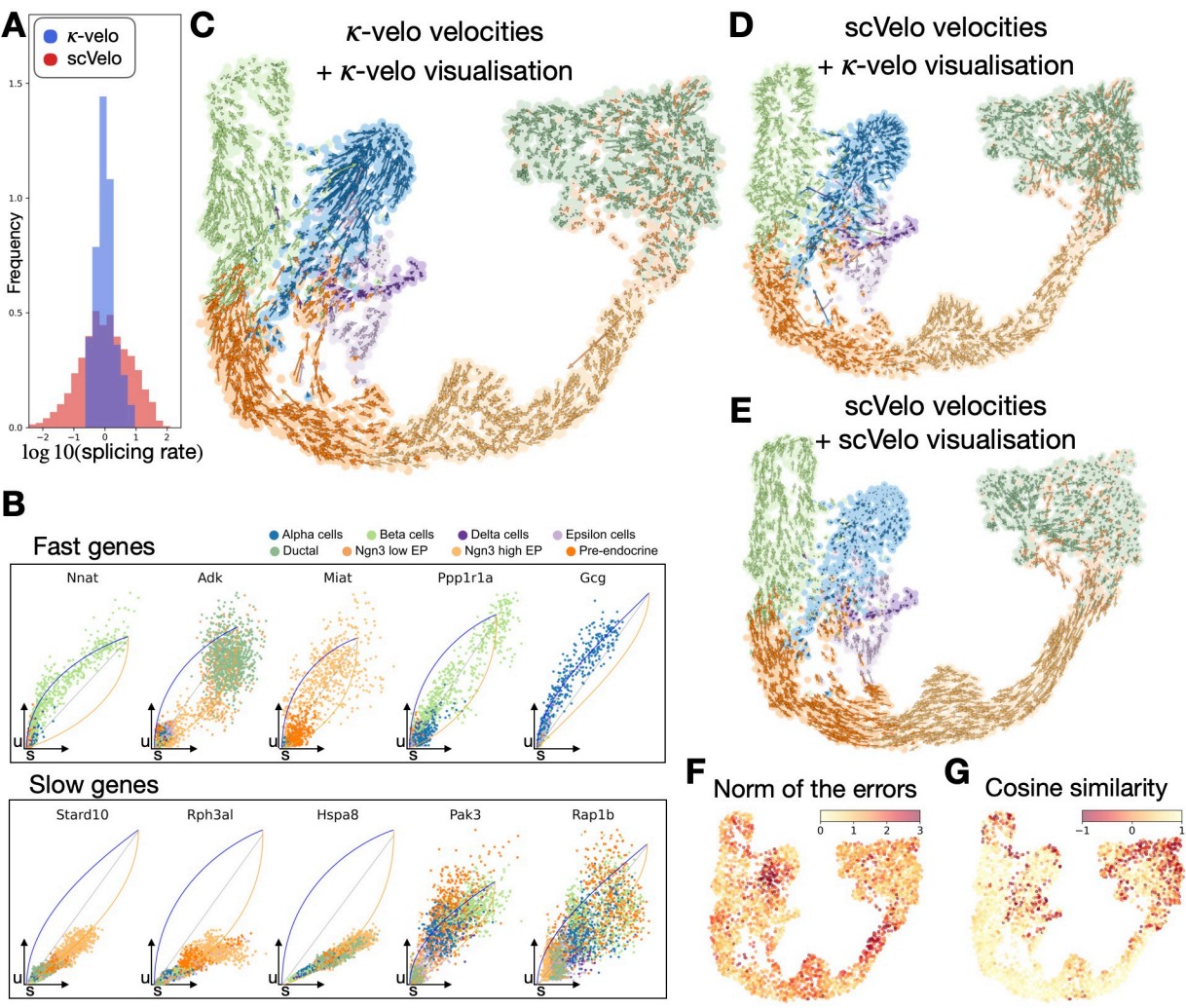

**Fig 4. κ-velo on pancreas endocrinogenesis. A**. Range of splicing rate $\beta$ estimated by scVelo (in red) and κ-velo (in blue). **B**. Examples of fast and slow genes, selected according to $\kappa\beta$. Learned kinetics are shown by blue (upregulation) and orange (downregulation) curves. **C**. Velocities from κ-velo projected onto a UMAP embedding using κ-velo projection. **D**. Velocities from scVelo projected onto the same UMAP embedding using κ-velo projection. **E**. Embedded velocities as returned by scVelo. For ease of comparison, plotting style was matched to (C) and (D). **F-G**. Quantitative comparison of the projected velocities from κ-velo (A) and scVelo (E) on the low dimensional embedding. We return the norm of the errors in **F** and the cosine similarity in **G**.

strong disagreement in the delta cells, which scVelo predicts to differentiate into the alpha cells, as well as in the alpha cells themselves that are predicted to have very small velocities all along the branch. Looking at single genes u-s phase portrait such as the *Gcg* gene (Fig 4B), we see that the cells are still differentiating and the full alpha branch has not reached the terminal state yet.

## κ-velo recovers multiple differentiation paths in hematopoietic system

RNA velocity analysis of single-cell datasets of differentiation of hematopoietic stem cells into different blood progenitor cells has proved difficult in the past [6, 7], and often the predicted velocities display a direction reversal. This reversal was attributed to genes with more complex kinetics leading to u-s phase portraits that do not have the shape expected from the current RNA velocity model. To investigate the potential of κ-velo on more complex datasets, we

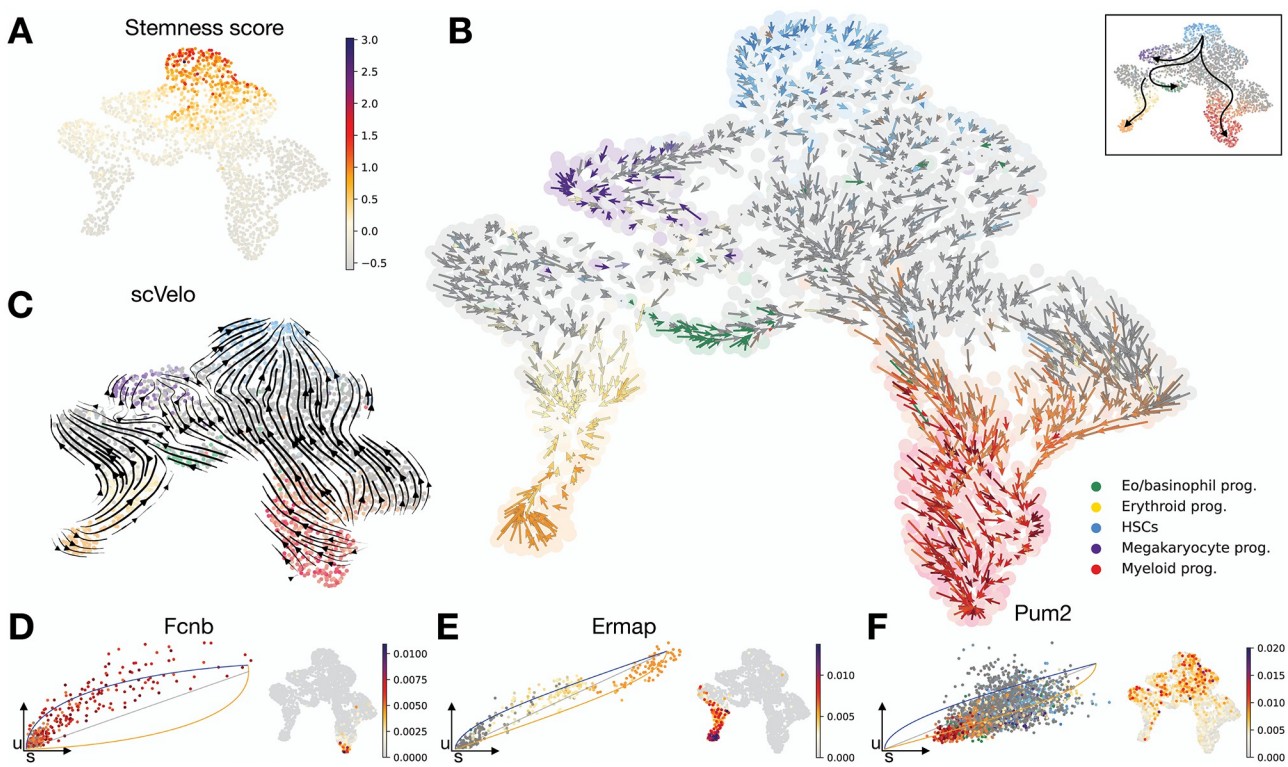

**Fig 5. κ-velo on hematopoiesis. A**. UMAP embedding with cells coloured for stemness score. **B**. κ-velo-recovered velocities projected onto UMAP embedding of the cells using Nyström projection. **C**. Velocities from scVelo projected onto the same UMAP embedding plotted using scVelo's velocity stream plot. **D-F**. Recovered dynamics in u-s portrait and expression UMAP of two fast genes *Fcnb* in **D** and *Ermap* in **E** and one slow gene *Pum2* in **F**.

applied the method to a dataset of murine hematapoietic stem and progenitor cells (HSPCs) [23]. The HSPCs in this dataset were acquired by sorting bone marrow cells using a broad Lin^neg c-Kit^+ (LK) gating strategy. Additionally, the datasets has been enriched for long-term hematopoietic stem cells (HSCs), which are usually less abundant than other populations. HSCs, which have a high multipotent potential (as indicated by the stemness score, Fig 5A) are at the beginning of the differentiation trajectory, and give rise to all mature blood cells [26]. In this dataset, these final states of mature blood cells are not yet reached since only HSPCs were included. Using a curated set of cell type gene markers, we identify the HSCs and progenitor populations, matching the original annotations (S18 Fig) [23].

The κ-velo pipeline correctly recovers the overall differentiation paths from the HSCs to various progenitor populations, such as to the myeloid and megakaryocyte progenitors (Fig 5B), while still capturing cell specific velocity variations (S19 Fig shows smoothed velocities on the embedding). The velocities show higher plasticity in the regions with higher stemness score and more commitment towards the ends of the differentiation branches. On the same dataset, scVelo recovers velocities in the exact opposite directions with velocities pointing from the more differentiated progenitor cells towards the HSCs (Fig 5C). We also identify fast genes, such as *Fcnb* and *Ermap* (Fig 5D and 5E), which are known to be involved in the commitment to the myeloid lineage and erythroid lineage respectively [27, 28]. *Pum2* is identified as a slow gene because its downregulation takes place over the full span from stem cell to progenitor (Fig 5F). This gene is known to suppress differentiation in HSCs [29].

## Eco-velo approximates cell state velocities using minimal data processing and computation

As a heuristic method that does not require cumbersome recovery of the rate parameters, we apply eco-velo on some of the introduced data sets. By simply taking the unspliced counts as a proxy of a cell's future state (Fig 6A), we can skip a few gene set filtering steps, imputation and parameter fitting, all of which are computationally expensive and can kill some of the true signal variability. We validate the model on a simulated dataset (Fig 6B and S20 Fig), where the model recovers the expected flow. We then test eco-velo on the pancreas endocrinogenesis dataset and the hematopoiesis dataset (pancreas endocrinogenesis: Fig 6C and S21 Fig for smoothed velocities, hematopoiesis dataset: S22 Fig). Since the method is based on the assumption that genes have the same splicing and degradation rates, and we know that cell cycle genes have different rates in the pancreas endocrinogenesis dataset, we exclude them from this analysis. The model delineates the directional flow from progenitor cells to alpha and beta cell fates. eco-velo also captures the high cell plasticities in the ductal population seen in Fig 4C. The final state of epsilon cells is also captured (S21 Fig smoothed) but the dynamics within the delta cells cannot be resolved. For delta and epsilon cells the issues could arise from trying to capture future states within sparse populations that are transcriptionally close to the more abundant population of alpha cells. A quantitative comparison of the projected velocities from eco-velo and $\kappa$-velo is shown in S23 Fig, where we see a strong similarity in the Ngn3 low endocrine progenitor, but more variation between the methods in the cycling ductal

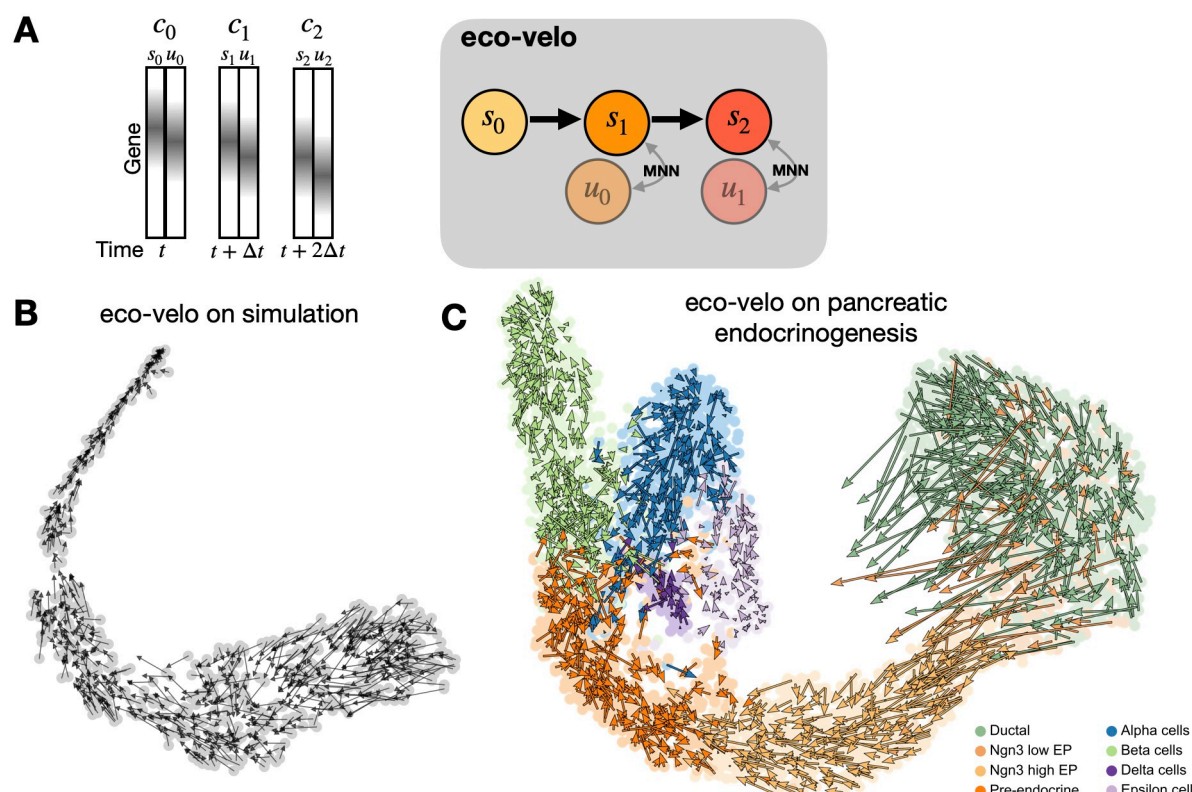

**Fig 6. Eco-velo as an alternative to computationally costly reaction rate parameter recovery. A**. Under certain conditions, a cell's unspliced state will represent the cell's future spliced state. To infer velocities, we look for the first MNN between a cell's unspliced counts and other cells' spliced counts. We draw an arrow from the cell to the identified MNN. **B**. We validate eco-velo on simulation and visualise the resulting velocities on t-SNE. **C**. eco-velo on pancreas endocrinogenesis.

population as well as in the terminal states. Given the strong theoretical assumptions of the model, eco-velo still captures the complex lineages of endocrinogenesis remarkably well. For the hematopoiesis dataset however, eco-velo is unable to capture the dynamics correctly (S22 Fig). Similarly to scVelo, the velocities are falsely projected back to the most stem-like state, hinting that the more basic assumptions about the splicing dynamics in eco-velo may not hold up for this particular biological process.

### Computational efficiency of the methods

We report the runtime on an Intel Core i5 CPU with 2GHz, 4 Cores and 16 GB of RAM. On the pancreatic endocrinogensis dataset with 3696 cells and top 5000 highly variable genes, the $\kappa$-velo workflow takes 15 minutes while the eco-velo worflow takes about 40 seconds. Full scVelo pipeline on the same dataset takes about 8 minutes.

### Data and software availability

All analysed datasets are publicly available. The pancreatic endocrinogenesis dataset is available from the Gene Expression Omnibus (GEO) under accession GSE132188 [20]. The murine gastrulation dataset is available on the Arrayexpress database (http://www.ebi.ac.uk/arrayexpress) under accession number E-MTAB-6967 [21]. For both datasets the count matrices can be downloaded directly from the scVelo Python implementation (https://scvelo.org) v0.2.4. The raw data from the chromaffin dataset is available on GEO under accession number GSE99933 [22]. The count matrices are made available by [1] at http://velocyto.org. The count matrices of the HSPC dataset are available on our GitHub Page: https://github.com/HaghverdiLab/velocity_notebooks. This GitHub page also contains all notebooks necessary to reproduce the results reported in this paper. A python implementation of the $\kappa$-velo and eco-velo pipeline can be found at https://github.com/HaghverdiLab/velocity_package.

## Discussion

In this manuscript, we study some of the current challenges in the inference of cell state velocities from scRNA-seq data and suggest novel approaches for tackling these problems. We argue that one of the interests in obtaining single cell velocities is to quantify the variation of dynamics among individual cells. This variance in single cell velocities can inform us about fluctuations of the dynamics, cell state plasticities and heterogeneity. We demonstrate that the processing procedure, several data smoothing steps and the visualisation approach in existing methods kill such biologically meaningful variance. The resulting information is closer to knowledge we could get from pseudotemporal ordering of cells than the true single cells velocity directions; one gets good looking cell velocity maps (i.e. conforming the expected pseudo-time directions) that do not reflect the reality of the information contained in the u-s mRNA data.

For applications in which obtaining the average cell state velocities over the specific time scale of mRNAs degradation is desired, we propose the eco-velo approach. It eliminates multiple cumbersome and error-prone steps, such as the gene-wise parameter estimation and visualisation of high-dimensional velocities.

For more detailed velocity analyses, we designed the $\kappa$-velo approach. The method recovers the full dynamics of splicing kinetics and addresses the relative scaling of velocity components across genes. We also design a consistent processing pipeline and suggest a new visualisation approach. We demonstrate how our model achieves better estimation of velocities than current methods on simulation. On real data, our method returns more plausible ranges of splicing rates and velocity magnitudes in several differentiation regions. $\kappa$-velo's velocity

components' scaling is based on the assumption that cell densities can be used as a proxy of typical travel time between two cell states. Heterogeneous cell birth and death rates along the differentiation path could partly disturb this assumption. To further improve this model, one could therefore consider estimating the heterogeneous cell birth and death rates based on the activity of apoptotic and proliferation genes [30]. Our results on simulation data (S8 Fig) demonstrate that the true global time of cells also resolves the scale-invariance issue. This indicates that other proxies of the true global time, e.g. cell density-scaled pseudotime, may also be used for inferring the relative scaling of velocity components among the genes in future work.

As described in Section "Careful processing prevents introduction of artefacts", we find that there can be difficulty in fitting reaction rate parameters for genes that do not display clear kinetic patterns of up- or downregulation on the u-s phase portrait. In the current version of $\kappa$-velo, we filter out genes where the fitted state assignments do not match the known pseudo-temporal order of cell types. In future work, we could use this prior information as initialisation in the parameter fitting procedure. The recovered high-dimensional velocity vectors now contain the deterministic part, but also capture the stochasticity of the dynamics. This can be used to perform several downstream analyses and answer questions about cell's progress through the dynamical process.

In the past, recovery of cell specific global latent time has been done after velocity analysis [2]. The recovery of a cell's global time was based on a heuristic integration of time assignment from individual genes. However, the gene-wise assignment of latent time are error-prone and additionally do not take into account the time that genes spend in steady-state. Integrating these errors does not necessarily mean that they cancel out. Because of these two reasons, recovery of global latent time should be done more carefully in follow up studies with strategies similar to CellRank [5], where several sequential cell state transitions are chained together to construct long transition paths along the differentiation manifold. Alternatively, estimation of global latent time may be integrated in the expectation maximisation procedure, similar to the approach in a recent preprint [11].

We also raise awareness about the time scales for which average velocities are being estimated. It would be interesting to measure velocities at multiple time scales to get an overview on the "plans" individual cells have in preparation for their short- or long-term developmental journey. One way of studying the changes that cells undergo at different time scales would be by inferring velocities from different sets of genes related to these time scales. For example investigating velocities on the time scale of the cell cycle or of the entire differentiation process. This also supports growing interest for inferring cell state velocities from other pairs of single-cell data modalities, e.g. mRNA coupled with protein levels [3], as they correspond to different time scales of gene regulation. Furthermore, inferring cell state velocities from modalities in which measurements are more accurate (in comparison to the uncertainty in quantification of unspliced-spliced mRNA counts) can enhance our ability to understand the biological variation in cell state velocities rather than variations due to measurement noise.

Estimation of cell state velocities in presence of multiple time point measurements or multiple batches of data collection is another important problem. However, the solution is not trivial as existing batch effect correction methods can distort the proportions between the s and u counts from separate batches. One possible strategy can be to estimate the velocities within each batch separately and visualise and project the estimated velocities on a shared embedding of all batches. Investigation of different approaches and possibilities remain open.

To conclude, we suggest that a comprehensive grasp of what we are actually estimating and visualising as cell state velocities is crucial for obtaining a full description of cell differentiation dynamics. True cell state velocities encompass both stochastic and deterministic parts of the biological dynamics. This information can be complementary to attempts for describing cell

differentiation as a full diffusion process [1, 5, 12, 15, 31–33] which contains the three terms of deterministic, stochastic and cell birth and death rates. Reliable quantification of cell state velocities in different transcriptional regions can put the relative magnitude (i.e. coefficients) of these terms into perspective in relation with one another.

## Supporting information

**S1 Table of contents. Table of contents of the main text.**
(PDF)

**S1 Appendix. Supplementary Notes A-J.** Details on theory, the algorithms, the simulation and processing of the data.
(PDF)

**S1 Fig. Average velocities for different time scales can be very different if the expression dynamics are not smooth.** On the left is the example of two noisy genes: the average velocity over $\Delta t_1$ is very different from the average velocity over $\Delta t_2$. For smooth gene dynamics as shown on the right, the average velocities are more similar.
(TIFF)

**S2 Fig. Density estimation for two simulated genes with different time scales.** $c = 10^{-3}$ is a constant scaling factor. The two simulated genes have the same reaction parameters $\theta$ but those for gene 2 are scaled by 10. (A) a slow gene, where no cells are in steady-state. The slope of the line gives us $\kappa_{g1}$ directly. (B) A fast gene, where a lot of cells are in steady-state. The slope of the red line gives us $\kappa_{g2}$.
(TIFF)

**S3 Fig. Comparison of recovery of scaling factors from unspliced counts (Eq 10) and from spliced counts (Note D in S1 Appendix).** (A) On simulation; the simulation is the same as in main Fig 3. (B) On the pancreas endocrinogenesis dataset.
(TIFF)

**S4 Fig. Overview of all processing steps in the $\kappa$-velo workflow.** In the middle, a schematic representation of how the spliced and unspliced matrices change during each step is shown. A size reduction of the coloured area indicates a filtering step where the number of genes are reduced. A change in colour represent a data manipulation, which does not changes the number of cells or genes, but changes the values in the matrix. On the left, some extra information is provided for some of the processing steps. More detailed information can be read in Note G in S1 Appendix. On the right, the u-s phase portraits of several example genes are shown to demonstrate how the different steps change the phase portraits, as well as which kind of genes are selected or removed in the filtering steps. Each of the genes is selected from the pancreas endocrinogenesis dataset that is analysed in main Fig 4.
(TIFF)

**S5 Fig. $\kappa$-velo and eco-velo applied on the chromaffin dataset.** The chromaffin dataset includes Schwann cell precursors (SCPs) (blue) differentiating into chromaffin cells (green). In the original paper, the purple cluster was identified as symphatoblasts and the yellow and red cluster as "bridge" cells [22]. (A) $\kappa$-velo applied on chromaffin dataset using PCA embedding for visualisation. Principal component (PC) 1 and 2 left and PC 2 and 3 right. (B) $\kappa$-velo applied on chromaffin dataset using UMAP embedding for visualisation (left: raw vector visualisation, right: smoothed vector visualisation). (A) and (B) show that $\kappa$-velo correctly captures the differentiation from SCPs into chromaffin cells. Interestingly, there also seems to be a

more committed differentiation in the bridge cells than the SCPs in the beginning of the manifold. (C) eco-velo applied on chromaffin dataset using UMAP embedding for visualisation (left: raw vector visualisation, right: smoothed vector visualisation).
(TIFF)

**S6 Fig. Projection of the velocity arrows (test set data points) onto existing embedding of initial cell positions (training set).** We compare our projection approach (left column) to scVelo's [2] (right column) projection for t-SNE [16] in (A) and (B) and UMAP [17] in (C) and (D).
(TIFF)

**S7 Fig. Projection of the velocity arrows (test set data points) onto existing diffusion map embedding of initial cell positions (training set).** We compare our projection approach in A to scVelo's [2]'s projection in B.
(TIFF)

**S8 Fig. Recovery of the scaling factor $\kappa$ from true time on simulation.** The simulation is the same as in main Fig 3. The factors are recovered similarly to the density approach described in Note C in S1 Appendix, except that $d(i, j)$ is calculated from $t_i$ the true simulated time of cell $i$: $d(i, j) = |(t_i - t_j)|$. Plotting $d$ on the $x$-axis and $f$ on the $y$-axis, the slope of the corresponding line gives us $\kappa$. Here, since we have true time, we do not need to exclude steady-states. (A) Comparison of the scaling factors recovered from true time to the true simulated factors. Note that here the range of recovered scaling factors is equivalent to the true factors because they were recovered from true time and not from a proxy of time that might be off by some constant factor. (B) Comparison of the factors recovered from the density approach to the factors recovered from true time.
(TIFF)

**S9 Fig. Comparison of the high-dimensional velocities recovered by $\kappa$-velo and scVelo on simulation for 100 genes with different speeds.** (A) High-dimensional velocity vector. One point represents a velocity for one cell for one gene. (B) We evaluate differences between true high-dimensional velocities and recovered velocities. We return the change in direction (cosine similarity), length (difference in vector norm) and the overall norm of the errors between real velocities and $\kappa$-velo velocities (in blue), or scVelo velocities (in red). To make the length comparable, the vectors high-dimensional vectors are normalised to have equal variance. Note the log-scale for frequency.
(TIFF)

**S10 Fig. Comparison of velocities recovered by $\kappa$-velo and scVelo on simulation projected on PCA embedding of spliced counts.** (A) Real simulated velocities (B) velocities recovered by $\kappa$-velo and (C) velocities recovered by scVelo projected on PCA. Cells on PCA coloured by norm of the errors between real velocities and (D) $\kappa$-velo velocities, or (E) scVelo velocities.
(TIFF)

**S11 Fig. Comparison of velocities recovered by $\kappa$-velo and scVelo on simulation projected on 2D-PCA embedding of spliced counts.** (A) Norm of the errors: $\|\vec{v}_t - \vec{v}_r\|$ with $\vec{v}_t$ the true 2D velocity vector on PCA and $\vec{v}_r$ the recovered vector. (B) Change in direction (cosine similarity) and length (difference in vector norm: $\|\vec{v}_t\| - \|\vec{v}_r\|$) for each cell in PCA space.
(TIFF)

**S12 Fig. The u-s phase portrait of *Acly*, *Dpysl2* and *Gnaz* (raw counts, after normalisation and after recovering of dynamics).** The u-s phase portrait of *Acly*, *Dpysl2* and *Gnaz* (from the

pancreas endocrinogenesis dataset), which are all genes with insufficient unspliced counts. Here, we show how scVelo would recover the dynamics if these genes were not filtered out. (TIFF)

**S13 Fig. Applying *κ*-velo processing pipeline on erythroid lineage dataset.** The scRNA-seq dataset on the erythroid lineage of mouse gastrulation [21] has been described in the context of RNA velocity by Barile et al. [6]. Here, we show that the subset has a varying ratio of total unspliced to total spliced counts in different cell types (A). This results in artefacts when using the standard scVelo processing pipeline (U and S normalised separately) (B, second row). Those artefacts are mostly resolved by normalising U and S combined (B, third row), which is part of the *κ*-velo processing workflow (B, last row). Using the *κ*-velo processing workflow fixes some of the reported de-differentiation (C). (TIFF)

**S14 Fig. Comparison of recovered reaction rate parameters on pancreas endocrinogenesis dataset.** Range of transcription rate $\alpha$, splicing rate $\beta$, and degradation rate $\gamma$ estimated by scVelo (in red) and *κ*-velo (in blue). (TIFF)

**S15 Fig. PCA projection of velocities in the pancreas endocrinogenesis dataset.** (A) Velocities returned by *κ*-velo projected on PCA embedding of spliced counts. (B) Velocities returned by scVelo projected on PCA embedding of spliced counts. We note that the gene space used is different for the two methods, as they have different criteria for gene selection. scVelo uses 1809 genes, while *κ*-velo uses 134. (TIFF)

**S16 Fig. Smoothed *κ*-velo projection of velocities in the pancreas endocrinogenesis dataset.** The two UMAPs compare (A) smoothed scVelo velocities projected by Nyström projection and (B) smoothed *κ*-velo velocities projected by Nyström projection. Velocities were smoothed by averaging over the 30 nearest neighbours. Neighbourhoods are calculated in S space. (TIFF)

**S17 Fig. Quantitative comparison of low-dimensional projection of velocities.** We compare scVelo velocities projected by scVelo $v_1$ to *κ*-velo velocities projected by Nyström-projection $v_2$ for every cell. (A) UMAP colored by cell types. (B) Difference in the norm of the two vectors $\|v_1\| - \|v_2\|$. (TIFF)

**S18 Fig. UMAP embedding of the HSPC dataset as calculated in the *κ*-velo pipeline.** Cells are coloured for (A) our assigned cell types (see Note I in S1 Appendix) or (B) the cell types assignments from the original data analysis [23]. (TIFF)

**S19 Fig. Smoothed *κ*-velo projection of velocities in the HSPC dataset.** Velocities were smoothed by averaging over the 30 nearest neighbours. Neighbourhoods are calculated in S space. Non-smoothed projection in main Fig 5B. (TIFF)

**S20 Fig. Eco-velo projection of velocities (calculated on simulations) shown on PCA in (A) and UMAP in (B).** (TIFF)

**S21 Fig. Smoothed eco-velo projection of velocities in the pancreas endocrinogenesis dataset.** Velocities were smoothed by averaging over the 50 nearest neighbours. Neighbourhoods are calculated in S space.
(TIFF)

**S22 Fig. Eco-velo applied on HSPC dataset using UMAP embedding for visualisation.** Left: raw vector visualisation, right: smoothed vector visualisation. Like scVelo (main Fig 5C), the velocities point from the more differentiated populations back to the stem cells.
(TIFF)

**S23 Fig. Quantitative comparison of low-dimensional projection of velocities.** We compare $\kappa$-velo velocities projected by Nyström-projection $\nu_1$ to eco-velo velocities projected onto the UMAP calculated in the $\kappa$-velo pipeline and shown in main Fig 4 for every cell. (A) UMAP colored by cell types. (B) Cosine similarity between the two vectors. (C) Norm of the difference between the two vectors $\|\vec{v}_1 - \vec{v}_2\|$. (D) Difference in the norm of the two vectors $\|\vec{v}_1\| - \|\vec{v}_2\|$. Cells are colored in grey when we do not have a velocity value for eco-velo, i.e. the cell does not have a mutual nearest neighbour within the top 50 neighbours.
(TIFF)

## Author Contributions

**Conceptualization:** Laleh Haghverdi.

**Data curation:** Brigitte Joanne Bouman, Yasmin Demerdash.

**Formal analysis:** Valérie Marot-Lassauzaie, Brigitte Joanne Bouman, Laleh Haghverdi.

**Funding acquisition:** Marieke Alida Gertruda Essers, Laleh Haghverdi.

**Investigation:** Valérie Marot-Lassauzaie, Brigitte Joanne Bouman, Fearghal Declan Donaghy, Laleh Haghverdi.

**Methodology:** Valérie Marot-Lassauzaie, Brigitte Joanne Bouman, Laleh Haghverdi.

**Resources:** Yasmin Demerdash, Marieke Alida Gertruda Essers.

**Software:** Valérie Marot-Lassauzaie, Brigitte Joanne Bouman.

**Supervision:** Marieke Alida Gertruda Essers, Laleh Haghverdi.

**Visualization:** Valérie Marot-Lassauzaie, Brigitte Joanne Bouman.

**Writing – original draft:** Valérie Marot-Lassauzaie, Brigitte Joanne Bouman, Laleh Haghverdi.

**Writing – review & editing:** Valérie Marot-Lassauzaie, Brigitte Joanne Bouman, Laleh Haghverdi.

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
