## [Decision Letter · Decision Letter 0]

19 May 2022

Dear Dr. Haghverdi,

Thank you very much for submitting your manuscript "Towards reliable quantifcation of cell state velocities" for consideration at PLOS Computational Biology.

As with all papers reviewed by the journal, your manuscript was reviewed by members of the editorial board and by several independent reviewers. In light of the reviews (below this email), we would like to invite the resubmission of a significantly-revised version that takes into account the reviewers' comments.

We cannot make any decision about publication until we have seen the revised manuscript and your response to the reviewers' comments. Your revised manuscript is also likely to be sent to reviewers for further evaluation.

Sincerely,

Wei Li, Ph.D.

Guest Editor

PLOS Computational Biology

Ilya Ioshikhes

Deputy Editor

PLOS Computational Biology

Reviewer's Responses to Questions

**Comments to the Authors:**

Reviewer #1: The review is attached as a docx file.

Reviewer #2: In this manuscript, the authors aimed to solve several key questions regarding the velocity estimation and visualization. They developed κ-velo which enables the estimation of the relative magnitude of velocity components across genes. At the same time, they developed a new method to visualize the velocity. Using both simulated and real data, the authors demonstrated that their method outperforms scVelo, one of the state-of-the-art methods for velocity estimation and visualization. The main comments and concerns are as follows:

1. The algorithm developed in this manuscript is not available, at least not user-friendly. Scripts in GitHub can only be used to generate figures for this manuscript. It is difficult for users to use this algorithm to analyze their internal data or other public data other than those mentioned in this manuscript.

2. In the manuscript, the authors address several key issues of velocity estimation and visualization in the Introduction section. All work in the manuscript is aimed at addressing these issues. Although these issues have been reported in detail in other studies, a brief description of these issues can help readers better understand this manuscript.

3. In Section 3.1, the authors demonstrate that their PCA embedding method can reliably represent high plasticity at the beginning and commit fate faster at the end of the trajectory through simulated data. However, with the nonlinear projection method, it appears that these phenomena cannot be observed. Does this mean that PCA-based methods and nonlinear projection methods can only show limited aspects of the velocity?

4. The operation of κ-velo does not take much computing resources and time. In this case, is eco-velo mode still necessary? Does this model have any other advantages or might be suitable for specific situations?

5. Are there any other datasets or cell types that can be used to test the performance of κ-velo? For example, single-cell data of different T-cell states.

Reviewer #3: See attached PDF.

**Have the authors made all data and (if applicable) computational code underlying the findings in their manuscript fully available?**

Reviewer #1: Yes

Reviewer #2: Yes

Reviewer #3: Yes

PLOS authors have the option to publish the peer review history of their article (what does this mean?). If published, this will include your full peer review and any attached files.

Reviewer #1: No

Reviewer #2: No

Reviewer #3: No
---

## [Decision Letter · Decision Letter 1]

26 Aug 2022

Dear Dr. Haghverdi,

We are pleased to inform you that your manuscript 'Towards reliable quantifcation of cell state velocities' has been provisionally accepted for publication in PLOS Computational Biology.

Best regards,

Wei Li, Ph.D.

Guest Editor

PLOS Computational Biology

Ilya Ioshikhes

Section Editor

PLOS Computational Biology

Reviewer's Responses to Questions

**Comments to the Authors:**

Reviewer #1: I believe that the authors have addressed all my concerns with either additional detailed analyses or appropriate theoretical clarifications. The limitations and possible improvements of current RNA velocity approaches that the authors pointed out will be valuable to the research community.

Reviewer #2: The authors have addressed all my questions and I have no further questions. I recommend accepting.

**Have the authors made all data and (if applicable) computational code underlying the findings in their manuscript fully available?**

Reviewer #1: Yes

Reviewer #2: Yes

PLOS authors have the option to publish the peer review history of their article (what does this mean?). If published, this will include your full peer review and any attached files.

Reviewer #1: No

Reviewer #2: No

---

## [Editor Report · Acceptance letter]

20 Sep 2022

PCOMPBIOL-D-22-00426R1 

Towards reliable quantification of cell state velocities

Dear Dr Haghverdi,

I am pleased to inform you that your manuscript has been formally accepted for publication in PLOS Computational Biology. Your manuscript is now with our production department and you will be notified of the publication date in due course.

With kind regards,

Zsofi Zombor
